# Short- and Long-Term Effects of a Prebiotic Intervention with Polyphenols Extracted from European Black Elderberry—Sustained Expansion of *Akkermansia* spp.

**DOI:** 10.3390/jpm12091479

**Published:** 2022-09-09

**Authors:** Simon Reider, Christina Watschinger, Julia Längle, Ulrike Pachmann, Nicole Przysiecki, Alexandra Pfister, Andreas Zollner, Herbert Tilg, Stephan Plattner, Alexander R. Moschen

**Affiliations:** 1Christian Doppler Laboratory for Mucosal Immunology, Faculty of Medicine, Johannes Kepler University, 4020 Linz, Austria; 2Department of Internal Medicine 2, Faculty of Medicine, Johannes Kepler University, 4020 Linz, Austria; 3VASCage (Research Centre on Vascular Ageing and Stroke) GmbH, 6020 Innsbruck, Austria; 4Division of Internal Medicine I (Gastroenterology, Hepatology, Endocrinology, and Metabolism), Department of Medicine, Medical University Innsbruck, 6020 Innsbruck, Austria; 5IPRONA AG/SPA, 39011 Lana, Italy

**Keywords:** black elderberry, polyphenols, prebiotic, microbiota

## Abstract

(1) Background: The intestinal microbiome has emerged as a central factor in human physiology and its alteration has been associated with disease. Therefore, great hopes are placed in microbiota-modulating strategies. Among various approaches, prebiotics, substrates with selective metabolization conferring a health benefit to the host, are promising candidates. Herein, we studied the prebiotic properties of a purified extract from European black elderberries, with a high and standardized content of polyphenols and anthocyanins. (2) Methods: The ELDERGUT trial represents a 9-week longitudinal intervention study divided into 3 distinct phases, namely a baseline, an intervention and a washout period, three weeks each. The intervention consisted of capsules containing 300 mg elderberry extract taken twice a day. Patient-reported outcomes and biosamples were collected weekly. Microbiome composition was assessed using 16S amplicon metagenomics. (3) Results: The supplementation was well tolerated. Microbiome trajectories were highly individualized with a profound shift in diversity indices immediately upon initiation and after termination of the compound. This was accompanied by corresponding changes in species abundance over time. Of particular interest, the relative abundance of *Akkermansia* spp. continued to increase in a subset of participants even beyond the supplementation period. Associations with participant metadata were detected.

## 1. Introduction

Although to date no distinct pattern of a healthy microbiota has emerged, certain alterations of microbiota composition and function are described as markers of disease-associated states. Collectively, these changes are summarized as dysbiosis, although a clear definition of the term is lacking and it has grown to be regarded increasingly unhelpful by experts in the field [1]. Nevertheless, loss of microbial diversity, a reduction in the number of beneficial microbes and an expansion of pathobionts are regarded as hallmarks of dysbiosis [2,3].

Because dysbiotic states of the microbiota are observed in and have been associated with numerous disease conditions [4], the field of microbiota modulation is receiving utmost attention. One potential means to influence this important determinant of health and disease are dietary interventions. The influence of long-term nutritional patterns on the microbiota has been studied and described extensively [5,6]. However, evidence suggests that drastic and sustained modifications of the diet are necessary to achieve long-term effects [7]. In a more focused approach, prebiotic supplementation could address this gap. The International Association for Probiotics and Prebiotics (ISAPP) defines prebiotics as nutritional compounds or substances which undergo selective metabolization by specific members of the intestinal microbiota, ultimately conferring a health benefit on the host [8]. While traditionally, non-digestible carbohydrates such as inulin, fructooligosaccharides and galactooligosaccharides have been considered prebiotics, the ISAPP has pointed out that some plant polyphenols might also fulfill these criteria and therefore might be considered prebiotics [8]. However, this is currently not reflected in the legal and regulatory status of these compounds within the European Union.

Polyphenols are a heterogenous class of secondary plant metabolites present in fruit and vegetables. Phenolic compounds possess one or more aromatic rings with two or more hydroxyl groups. They can be present in free forms or conjugated with sugars, acid and other biomolecules [9,10,11].

Plant polyphenols are not readily taken up in the small intestine and reach the colon in considerable amounts, where they can serve as substrate for the colonic bacterial microbiota [9]. As potential health effects of polyphenols seem to depend on the resulting metabolites from degradation by the colonic microbiota [12], the individual microbiome configuration seems to be a key determinant of potential effects of polyphenol-rich black elderberry extracts. The bioavailability of polyphenols is variable and depends on the nutritional setting in which the substance is presented and an extensive secondary metabolism, largely by the intestinal microbiome [13,14,15,16,17,18,19]. Thus, microbiome composition might influence polyphenol effects in given individuals through diverse modification [20,21]. Bacterial secondary metabolization of dietary polyphenols improves bioavailability and hydrophilicity of these compounds and enables detection in the urine [20]. One systematic review identified data hinting at potential antibacterial and anti-inflammatory effects, although underlying mechanisms remained elusive and the overall level of evidence was classified as weak [22]. Anthocyanins of black elderberry extract are taken up and incorporated into endothelial cell membranes and cytosol, hinting at potential antioxidative benefits [23]. On the other hand, microbiota-modifying properties of polyphenols have been reported and include an expansion of Bifidobacteria [24] and Lactobacilli [12,25] and alterations of the Bacteroides–Firmicutes ratio, possibly reflecting an adaption to a higher glycan load [26]. Besides direct prebiotic effects on the microbiota, an environmental mechanism has also been proposed for polyphenol-associated microbiota modulation. Certain polyphenols have been shown to exert anti-microbial activity in vitro [27]. Therefore, broad antimicrobial activity of certain polyphenol compounds could serve to open up niches in the gastrointestinal tract and allow for expansion of other taxa. To include this dualistic effect, the term “duplibiotics” has been coined [28,29].

Food-derived anthocyanins exhibit a low bioavailability in their native form [15] but are partly metabolized and taken up in the intestine and can be recovered from the urine [13,14,15,16,17,18,19,30,31].

Effectively, the polyphenol–microbiota axis exerts influence on host physiology. Certain polyphenol metabolites of microbial origin serve to strengthen barrier function and boost secretion of antimicrobial peptides [32] and effects on secondary bile acid metabolism have been described [33]. In a mouse model, food rich in polyphenols attenuated detrimental metabolic and inflammatory responses [34]. Short-chain fatty acids are also produced from polyphenols by the intestinal microbiota [35] and may serve as regulators of colonic mucosal homeostasis [36].

European black elderberries (Sambucus nigra) and their extracts have been used traditionally in parts of Europe for centuries as a remedy for common colds and influenza-like symptoms. Recently, an alleviating effect of an extract of black elderberries on symptoms and duration of influenza infections has been shown [37,38,39]. As research has mostly focused on the effects of black elderberry on upper respiratory tract infections, little is known on its microbiota-directed effects. Black elderberries are rich in polyphenols [27] and also contain various immunomodulating polysaccharides such as pectin [40,41]. The main group of polyphenols in black elderberry fruits are anthocyanins (cyanidin 3- sambubioside-5-glucoside, cyanidin 3,5-diglucoside, cyanidin 3-sambubioside, cyanidin 3-glucoside). Additional polyphenols present in black elderberry are chlorogenic acids, rutin, and isoquercitrin [42].

This study aimed to provide a comprehensive in depth characterization of the microbiota-modulating effects of a 3-week intervention with black elderberry extract, standardized to anthocyanin and polyphenol levels.

## 2. Materials and Methods

### 2.1. Trial Design and Study Cohort

The ELDERGUT trial was a 9-week longitudinal intervention study with 3 distinct study phases of 3-week duration each. A baseline phase serving to obtain robust initial measurements was followed by a 3-week intervention period during which a daily dose of 600 mg of highly purified black elderberry extract (300 mg twice daily) was consumed by the participants. Participants logged digestive symptoms, bowel movements and possible adverse effects in a study diary provided by the investigational team. Participants could be either male or female and had to be 18 to 50 years old. These individuals had to report a history of vaginal delivery and being breast-fed for at least 3 months after birth. Exclusion criteria were recent antibiotic treatment within 3 months of the beginning of the study, a history of gastrointestinal disease or other acute or chronic medical conditions with the potential to affect the intestinal microbiome, a history of major abdominal surgery (appendectomy excluded), baseline consumption of dietary supplements or probiotics, a strictly vegan diet, relevant pathologic abnormalities in baseline clinical biochemistry tests and a known allergy to black elderberry.

The ELDERGUT trial (Figure 1A,B) included n = 30 healthy individuals (15 females, 15 males) that volunteered to participate in this study. All study subjects completed the 9-week study period and provided biological samples and clinical information at weekly study visits as indicated in Figure 1A. No study drop-outs occurred. All patients underwent a thorough clinical and biochemical evaluation at the baseline. Fecal calprotectin—a marker of intestinal inflammation—was within the normal range in all study subjects. Notably, two study participants happened to have a positive screening test for IgA antibodies against tissue transglutaminase and the diagnosis of celiac disease was confirmed histologically in both participants. These were excluded from the analysis as celiac disease was a pre-defined exclusion criterion. During conduction of the trial, 2 male participants contracted SARS-CoV2 during week 5 of the study period. The disease course was mild and the infection did not result in exclusion from the study or patients not providing samples.

The mean age of the trial population was 23.7 years (±3.05 years, range 19–31 years) and the mean body mass index was 22.3 kg/m^2^ (±2.49, range 17.7–27.3). Baseline questionnaire-based nutritional assessment revealed a well-balanced nutritional pattern with differences between female and male participants. Male participants reported less intake of dietary fiber (*p* < 0.01, chi-square test), and a considerable but not significantly larger proportion of daily intake of animal fat (Figure 1C).

### 2.2. Intervention

Capsules containing 300 mg European black elderberry extract (ElderCraft^®^) were provided by IPRONA AG/SPA (Lana, Italy). ElderCraft^®^ is a full spectrum water extract standardized to 14% anthocyanins (primarily cyanidin 3-sambubioside-5-glucoside, cyanidin 3,5-diglucoside, cyanidin 3-sambubioside, cyanidin 3-glucosid) and 18% polyphenols (primarily chlorogenic acids, rutin, isoquercitrin).

### 2.3. Biosample Acquisition and Biobanking

During the trial period, weekly biospecimen sampling (feces, urine, saliva) was performed. Additionally, blood samples were drawn once per study phase (at weeks 1, 6 and 9). Fecal samples were collected simultaneously into tubes pre-filled with nucleic acid stabilizing solution (INVITEK, Berlin, Germany) and standard fecal collection tubes (Sarstedt, Nümbrecht, Germany). Stabilized fecal samples were stored at −20 °C, unstabilized samples were stored in aliquots at −80 °C. Urine samples were centrifuged and the supernatant stored at −80 °C. Saliva samples were directly stored at −80 °C. Blood samples were collected into EDTA- and Serum-Tubes (Sarstedt, Nümbrecht, Germany), centrifuged and the supernatants were stored at −80 °C.

### 2.4. Fecal DNA Extraction

Fecal DNA was extracted from stabilized samples, according to the instructions provided by the manufacturer using a spin-column based kit (Qiagen Fast DNA Stool Kit, Qiagen, Germany). Briefly, homogenized suspensions were heated to 95 °C for 10 min, centrifuged and treated with Proteinase K. Then, samples were incubated with the buffer provided in the kit and pure ethanol. After vortexing, the preparations were transferred onto spin columns and centrifuged, followed by multiple washing steps on the column using the washing buffer from the isolation kit. Finally, DNA was eluated in 100 µL of a TE-based buffer. Nucleic acid concentration was measured using a Nanodrop 1000 spectrophotometer (Eppendorf, Hamburg, Germany).

### 2.5. Metagenomic Analysis

Fecal DNA isolates were analyzed using 16S amplicon based metagenomics in cooperation with a commercial provider. DNA concentration and purity was monitored on 1% agarose gels and DNA was amplified using a PCR with primers targeting the V3-V4 variable regions of the bacterial 16S rRNA gene (primer pair 314F-806R) All PCR reactions were carried out with Phusion^®^ High-Fidelity PCR Master Mix (New England Biolabs). PCR products were mixed at equal ratios and purified using Qiagen Gel extraction kit (Qiagen, Germany). Libraries were prepared using the NEBNext Ultra DNA Library Prep Kit for Illumina and quantification was done using Qubit and qPCR. Sequencing was performed on the Illumina Miseq platform with PE250 chemistry.

Paired end reads were assigned to samples using specific barcodes added in the PCR amplification step. Barcode sequences were removed and reads were merged using FLASH (v.1.2.7; Baltimore, MD, USA) [43]. Resulting raw tags were quality filtered using QIIME (v.1.7, Boulder, CO, USA [44]). Chimera detection and removal was performed using UCHIME (Boulder, CO, USA) against its Gold reference database [45,46]. OTU clustering was performed using UPARSE version 7 [47] using effective tags at a clustering level of 97%. Representative sequences were annotated for every OTU using mothur [48] and the SILVA database [49]. Further downstream analysis including alpha- and beta-diversity and analysis of differential abundance is detailed in the statistics section.

### 2.6. Statistics

Data from study diaries were summarized using descriptive statistics and significance was tested using ANOVA with the post hoc Tukey test and the Kruskal–Wallis/Wilcoxon test for continuous numeric variables depending on the underlying distribution. The frequency of symptoms and data from the nutrition questionnaire was assessed using chi-square statistics. Amplicon-based metagenomic data were imported into R v. 4.1.2 (R Core Team, Vienna, Austria) [50] and RStudio using phyloseq [51] and then analyzed according to published workflows. Permutational analysis of variance (PERMANOVA) was performed using Vegan (v.2.5-7, Helsinki, Finland) [52]. Trends in differential abundance of bacterial species over the duration of the study period were explored using a linear mixed model implemented in Maaslin2 [53]. For these approaches, appropriate variance and occurrence filters were implemented (Appendix A and analysis scripts are provided). The tidyverse [54], rstatix [55] and ggpubr [56] packages were used for data cleaning and statistical procedures.

## 3. Results

### 3.1. Patient-Reported Outcomes during the Trial Period

Participants completed symptom-based questionnaires on a weekly basis, assessing typical gastrointestinal symptoms as well as measures of general health and well-being. Additionally, information on the number of bowel movements and their consistency (Bristol Stool Scale, BSS value) was recorded.

Overall perception of health and quality of life of the study participants was consistently within the upper third of the 7-point visual analogue scale used for assessment. No dynamic was observed along baseline, intervention and washout phase of the trial. Likewise, no differences between female and male participants were observed within these parameters (Figure 2A).

Study subjects were asked for the presence of gastrointestinal symptoms including increase or decrease in appetite, the occurrence of flatulence and diarrhea, nausea, constipation and abdominal discomfort. Of these, only flatulence was a regular complaint, but no clear association with the intervention emerged. Abdominal pain was slightly more severe and more frequent in female participants, although in most instances this complaint coincided with specific events of the menstrual cycle. Overall, no symptom-based signal associated with the intervention was observed (Figure 2B).

The number of bowel openings was recorded and calculated as mean number of bowel movements per week. No changes were observed on a week-to-week basis, particularly not within the intervention period (Table 1, Figure 2C).

The Bristol Stool Scale (BSS) is a widely-used and validated tool [57,58] to assess the consistency of a bowel movement. Higher values on the BSS indicate more liquid stool, low values represent very hard consistency. Similar to bowel movements, the mean weekly BSS value (mean of the BSS value of all bowel movements of a single study participant during every week of the trial) was calculated and happened to remain in the middle part of the scale and unchanged during the whole trial period (Table 1, Figure 2D).

### 3.2. Alterations in 16S Amplicon-Based Metagenomic Profiles at the Beginning and after the End of the Prebiotic Intervention

A 16S amplicon based metagenomic approach was applied to investigate temporal alterations in microbiota composition associated with an intervention with ElderCraft^®^. Weekly fecal samples from 28 participants were analyzed (252 samples overall) and the resulting profiles were compared on a weekly basis to the baseline configuration, i.e., median abundances from weeks 1 to 3.

### 3.3. Changes in Intra-Sample Diversity

In the first week of the intervention period an immediate and strong increase in measures of α-diversity (i.e., intra-sample diversity) was observed (Figure 3A). The number of observed species increased from 623.4 ± 58.0 (mean and standard deviation) during baseline to 925.3 ± 348.2 at the end of week 4. Similarly, the Shannon index increased from 4.1 ± 0.27 to 4.45 ± 0.49 (*p* < 0.001 for both comparisons; pairwise Wilcoxon test). The most prevalent OTUs were mapped to the genus *Bacteroides* (phylum *Bacteroidetes*) and *Faecalibacterium* (phylum *Firmicutes*). Additionally, other genera from the phylum *Firmicutes* were highly abundant both at baseline and throughout the study period (i.e., *Agathobacter*, *Ruminococcus*, *Roseburia*). This was also observed for the genenera *Bifidobacterium* (phylum *Actinobacteria*) and *Akkermansia* (phylum *Verrucomicrobia*).

In the later weeks of the intervention period, namely weeks 5 and 6, measures of α-diversity decreased but remained significantly higher in terms of the number of observed species (818.5 ± 356.6, 629.8 ± 43.0). The Shannon index was significantly different to baseline only were not significantly different from baseline apart from week 4 (Figure 3A). After stopping the prebiotic intervention in week 7, again an increase in the number of observed species was evident (897.9 ± 228.9, *p* < 0.001, pairwise Wilcoxon test). There was no significant difference of Shannon indices.

### 3.4. Compositional Differences over Time

Differences of overall microbial composition between the different study weeks were investigated using the unweighted unifrac index of α-diversity as a measure of difference between groups of samples (Figure 3B,C). Both paired data and principle coordinate analysis (PCoA) revealed changes during, most pronounced in weeks 4 and 5, and in the first week after the prebiotic intervention. These differences met the prespecified significance level. Additionally, a significant effect of sex on microbiome composition was evident (*p* < 0.01 for both timepoint and sex, R-squared 0.07 for timepoint and 0.02 for sex, permutational analysis of variance).

### 3.5. Alterations on a Species Level Driving Overall Changes in Bacterial Ecology

Next, we employed linear modelling—taking the longitudinal nature of our data into account—to identify differences in the abundance at the species level responsible for driving the observed microbial ecologic dynamics.

Overall, on the genus level 36 genera and on the species level 71 differently abundant microbial taxa were detected when compared to the baseline abundances (q value < 0.05, Maaslin2 linear random mixed-effects model, Appendix A). The taxa identified could be attributed to all major microbial phyla of the human fecal microbiota. Confirming findings from diversity analyses (Figure 3), most species exhibited significantly different abundance in week 4 and week 7 compared to baseline abundances, i.e., at the beginning of the intervention and washout periods, respectively. Most of the detected species showed transient increases such as *Butyricicoccus* spp., Fusobacterium mortiferum, certain species of Ruminococci or decreases including different Roseburia species and Bifidobacterium adolescentis in response to the supply or withdrawal of black elderberry extract. However, some taxa including Akkermannsia were found to have a sustainable expansion throughout the washout period. The most relevant genera and species identified are outlines in Figure 4A,B and a summary statistic for relative abundances as well as information on significance levels derived from Maaslin2 are shown in Table 1.

### 3.6. Factors Determining Response to the Intervention

Considerable interest is directed towards inter-individual differences related to differential effects of prebiotic interventions. We therefore associated baseline demographic, behavioral and nutritional covariates with increases in *Akkermannsia* spp. of at least 3-fold during the trial period (i.e., 10 out of 28 participants). Daily intake of foods rich in plant fat was associated with a relevant increase in the abundance of *Akkermansia* spp. in response to the intervention (*p* = 0.03, Fisher’s exact test). However, no associations of baseline α- or β-diversity with such an increase were detected. Likewise, we detected no baseline microbial signature predicting a relevant increase in *Akkermansia* abundance.

An increase in the number of observed species or in Shannon index values at week 4 and week 7 of the study compared to baseline was considered a second marker of individual response to the prebiotic. For the number of observed species, this was defined as an increase by at least 50% while for Shannon indices the 3rd quartile value (1.1687 for week 4 and 1.0568) was applied. Thus, based on observed species count, 17 and 8 participants were classified as α-diversity-responders at week 4 and week 7, corresponding to 60.7% and 28.6% of participants. Regarding Shannon index cutoffs, seven individuals were classified as α-diversity-responders at both timepoints (25%). Univariate correlation with baseline nutritional patterns revealed a significant association of α-diversity-response at week 4 with reduced consumption of dairy products such as cheese, eggs and milk (for Shannon index values) and rice (based on observed species counts). Increases in α-diversity at week 7 were associated with less baseline consumption of whole grains (Shannon) and pasta (observed species count). However, there was no association of β-diversity response at weeks 4 and/or 7 with baseline microbiome composition (i.e., Bray–Curtis dissimilarity indices comparing β-diversity responders at baseline) and specific bacterial taxa on the genus and species level (linear mixed effect model Maaslin2; data not shown).

## 4. Discussion

Prebiotic interventions are an attractive and promising platform towards modification of the intestinal microbiome. Their comparatively broad effect on different, albeit rather specific bacterial strains, holds the potential to provide symptom relief yet a potentially mechanistic approach in microbiome-mediated diseases [8,59]. To better understand the interactions of a polyphenol-containing prebiotic based on black a elderberry extract, the ELDERGUT trial has been designed and performed. The design of this 9-week longitudinal trial with a sequence of baseline, intervention and washout periods allowed for assessment of intra-individual dynamics and mitigates the considerable baseline variation that is a common characteristic of studies of the intestinal microbiome [60,61].

Dietary polyphenols have multifaceted and often diverging effects on members of the intestinal microbiota. Extensive metabolization by the microbial enzymatic machinery results in active secondary metabolites [12,15,35]. The aim of the ELDERGUT trial was to investigate the prebiotic properties of a phenol-rich elderberry extract. This highly purified preparation enables a polyphenol content of 18% a dose corresponding to a daily consumption of 30 g raw fruit.

In our population the intake of this concentration of black elderberry extract did not result in any adverse effects. No changes in overall perception of health and quality of life as an array of digestion-associated symptoms was observed. This confirms the good tolerability of the substance which is a prerequisite for further investigation as a prebiotic. Notably, no effects on fecal consistency and the number of bowel openings were observed. These are a typical symptoms induced by many other prebiotics through osmotic and bulk-forming effects [8,62,63].

Although no effects on patient-reported outcomes were detected, the introduction and withdrawal of the study medication at week 4 and week 7 of the trial resulted in changes of measures of microbial α- and β-diversity. As a measure of within-sample variability, indices of α-diversity (observed species count, Shannon index) spiked in these moments of perturbation. Between sample variability, i.e., β-diversity, was quantified using Bray–Curtis dissimilarity. Primary coordinate analysis and PERMANOVA statistical analysis revealed significant, although comparably small, associations with sex and week of the trial, the latter corresponding to the status of supplementation. Overall, 92 bacterial taxa were identified at the species level to be differentially abundant in fecal samples when comparing the weeks during the intervention and/or washout period with the median abundance at baseline. As sudden changes in diversity measures indicate perturbation of microbial community structures this points toward a relevant microbiome-shaping capacity of this prebiotic intervention. Interestingly however, these changes in diversity were only observed immediately after the start and end of the intervention period, respectively. This finding could reflect a quick stabilizing response within the microbial communities. In the context of well-known properties of the intestinal microbiome [7], this observation is likely to reflect the resilience and stability of a healthy microbiome configuration in this very uniform study population of student volunteers. This also underscores that the stability of microbiome compositions might result in obstacles to all variations of microbiome-targeted therapies, underlining the relevance of individual baseline composition and microbiome properties for prebiotic effects.

As the study population in the ELDERGUT trial included only healthy young volunteers, applicability of these findings in the general population and especially in certain disease groups such as patients suffering from inflammatory bowel disease or irritable bowel syndrome might be limited. Overall quality of life and subjective perception of health were already high at baseline in this cohort. Therefore, there is a possibility of a beneficial effect of black elderberry extract on these measures that could not be detected in this trial. Furthermore, minor adverse events which could be potentially connected to the intervention are difficult to validate in the absence of a true placebo control group and without adequate statistical power to address this issue. Another possible limitation is, that the study intervention consisted of a natural, although highly purified mixture of different polyphenols and cannot be considered a chemically well-defined pure substance. Therefore, attribution of effects to single compounds is not possible.

Statistical methods to assess differential abundance in microbiome studies are notoriously non-robust, with different methods resulting in often differing and even contradicting conclusions. These discrepancies can depend on choices made during processing of the data and intrinsic properties of the dataset. Furthermore, some methods which have been used in the field have been reported to suffer from poor specificity and a broadly accepted ‘state of the art’ workflow is lacking [64,65]. Nevertheless, recent studies have made the effort to systematically compare the different methods that are currently in use [64]. On the basis of these published findings, we chose a method that has been scoring consistently in these artificial settings and also conceptionally addresses the nature of this specific dataset [53]. Thus, we are confident, that the taxa that are identified in the present work represent true hits with biologic significance. In line with earlier reports [66], we detected a significant increase in the abundance of *Akkermansia* spp. associated with the study intervention. This bacterial taxon is known for its beneficial effects on inflammation and metabolism [67,68]. It can influence and strengthen the intestinal barrier despite its mucolytic properties. Notably, *Akkermansia* has already been reported to bloom under certain polyphenol-rich diets, although no polyphenol degrading enzymatic capabilities of this genus could be delineated. It is assumed, that this effect could be due to indirect interactions between dietary polyphenols and *Akkermansia* [25,28,35], mediated by other members of the microbiota. Whether this specific effect can be considered truly prebiotic by the most recent ISAPP definitions is controversial [28]. The bacterial genus Suterella from the phylum Proteobacteria was significantly increased immediately in the first week of supplementation, although it returned to values non-significantly different from the baseline during the intervention period. This taxon has been beneficially associated with glucose metabolism and metabolic changes after Roux-Y-gastric bypass [69]. As members of this genus possess epithelium-adhering and mild pro-inflammatory properties without causing barrier disruption, it has been postulated, that this bacterial group might exert immunomodulatory effects [70]. On the other hand, in studies on fecal microbiota transplantation for Ulcerative Colitis, prevalence of this genus in the fecal transplant has been associated with a lack of remission after the procedure [71]. This detrimental effect on intestinal inflammation was linked to its ability to degrade host IgA [72].

Within the genus Bacteroides (Phylum Bacteroidetes) a more nuanced effect was detected with granular differential effects at the species level. Bacteroides cellulosolyticus increased early during the intervention. This carbohydrate-degrading species [73] has been associated with plant based nutrition [74], better outcomes of anti PD-L1 immuno-oncologic therapy [75] and disease course in COVID-19 [76]. Furthermore, Bacteroides thetaiotaomicron [77] has been implicated in the context of inflammatory bowel diseases [78,79] immune system maturation and obesity [80] and is well adapted to its existence as a gut commensal [81]. The genus Bacteroides seems to emerge as a key target genus of polyphenol supplementation in the setting of this study, probably reflecting the extensive enzymatic repertoire encoded by this bacterial taxon.

Observing the high inter-individual variability of changes in abundance during the intervention, we hypothesized that there might be underlying differences between participants at baseline that could be used to predict probability of taxonomic changes during supplementation. We identified an increase in *Akkermansia* spp. as most relevant to host health and tested for associations of baseline demographic (i.e., sex) and nutritional variables with a 3x increase in *Akkermansia* by week 4. This analysis revealed a significant association of an increase in *Akkermansia* with daily consumption of plant fats. Of note, baseline microbiome diversity indices and taxonomic composition were not significantly associated with this increase in *Akkermansia* abundance. Likewise, we detected associations of changes in α-diversity upon supplementation of black elderberry extract with baseline nutritional features but not with a distinct microbial signature at the baseline. Nevertheless, further analyses could provide better mechanistical insights in the interaction of dietary polyphenol intake, microbial abundance, enzymatic functions within the intestinal microbiota and resulting effects on the host.

## 5. Conclusions

The ELDERGUT trial provided convincing evidence for the prebiotic properties of a polyphenol-rich black elderberry extract in healthy individuals. While intake of this extract was associated with specific and individualized taxonomic changes within the fecal microbiota, no convincing associations with baseline factors emerged in this analysis.

## Figures and Tables

**Figure 1 jpm-12-01479-f001:**
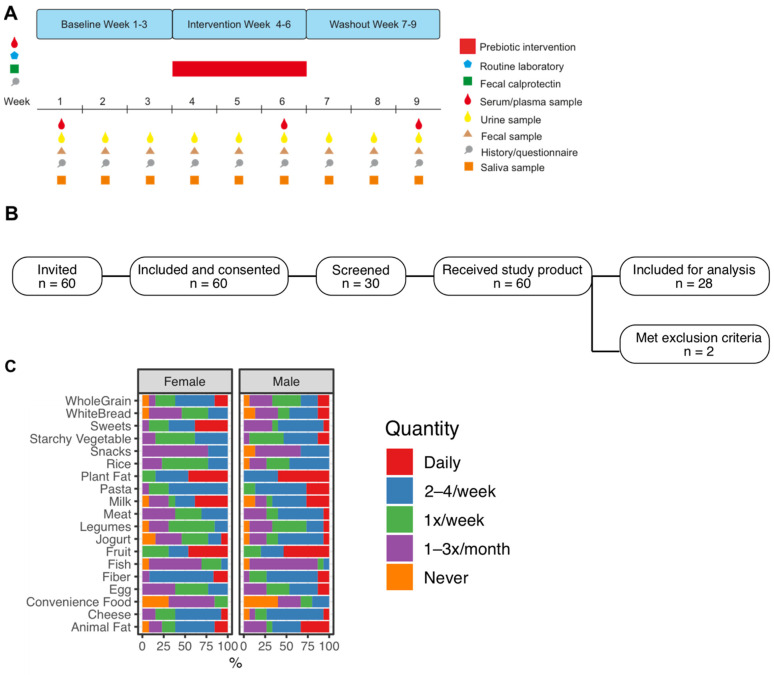
ELDERGUT trial design and baseline nutrition data: (**A**) Timeline of interventions and sampling during the 9-week study period, depicting the baseline, the intervention and the washout phase of the study. During the study, biological samples including urine, faeces, and saliva were collected once a week, blood samples were drawn once per study phase (i.e., in weeks 1, 6 and 9). Questionnaires on well-being and gastrointestinal symptoms were completed by the participants during weekly visits. (**B**) Overview of the inclusion and screening strategy. (**C**) Data from questionnaires at baseline, indicating nutritional patterns within the cohort as well as gender-associated differences in food intake.

**Figure 2 jpm-12-01479-f002:**
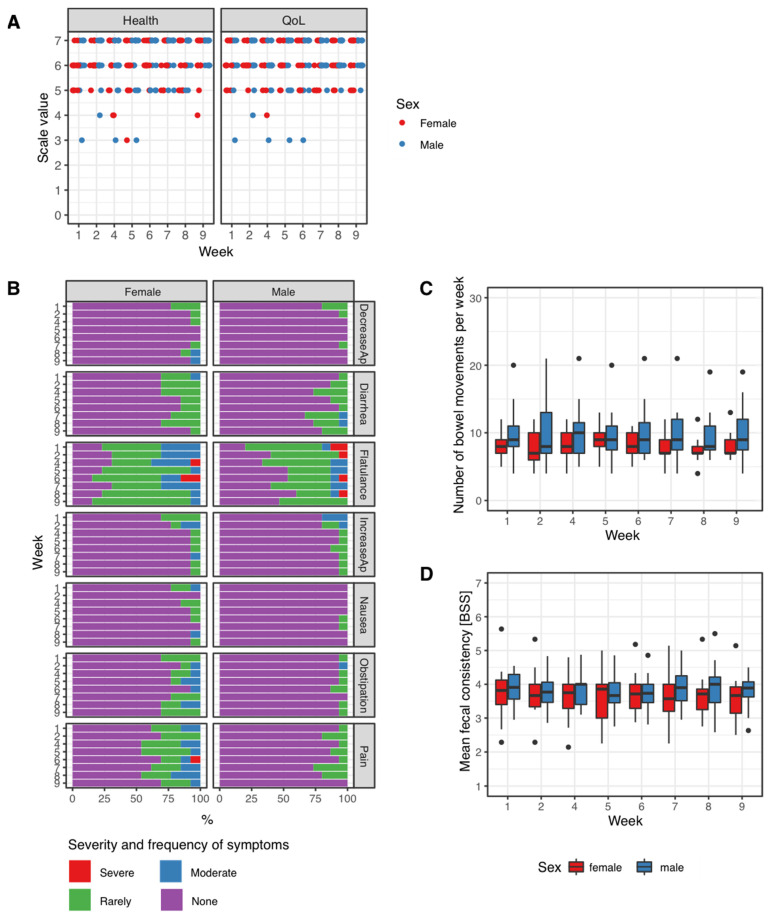
Patient-reported outcomes. (**A**) Weekly scores for health and quality of life (QoL), (**B**) results from weekly symptom-based questionnaires including common gastrointestinal symptoms. (**C**) Number of bowel movements per week in female and male participants. (**D**) Mean fecal consistency, i.e., the Bristol Stool Scale (BSS) value per week. Data were compared by the Kruskal–Wallis test, no significant differences were observed.

**Figure 3 jpm-12-01479-f003:**
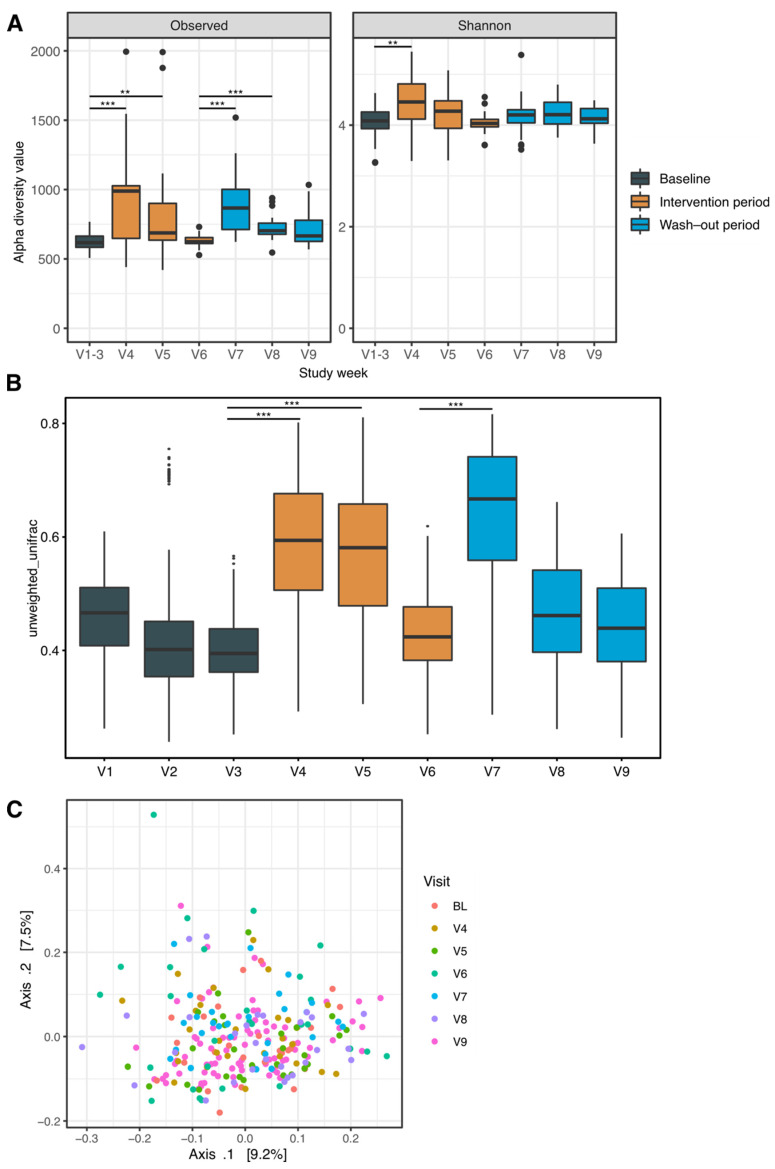
Microbial ecology measures of diversity. (**A**) α-diversity—describing intra-group diversity, i.e., the number of observed species—and Shannon indices. (**B**,**C**) β-diversity as unweighted Unifrac values by study week (**B**) and a Principal Coordinate Analysis of unweighted Unifrac distances (**C**). Statistical comparisons were made using pairwise Wilcoxon tests between the median of the baseline values (weeks 1–3) and the subsequent weeks. *p* values ** < 0.01, *** < 0.001.

**Figure 4 jpm-12-01479-f004:**
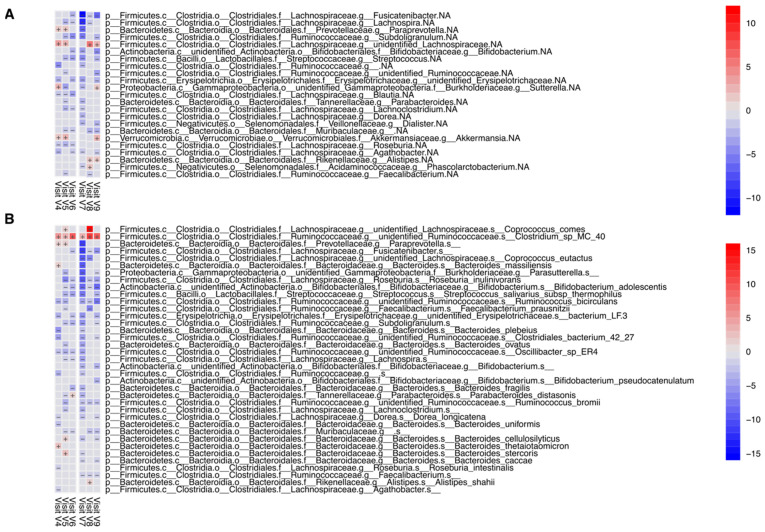
Analysis for differential abundance of bacterial taxa at the genus (**A**) and species (**B**) levels. Blue and red squares depict coefficients of significantly differential abundance for every week of the trial, grey squares represent non-significance (Maaslin2 linear mixed model, *q*-value < 0.25, *p* value < 0.05; detailed results provided in Appendix A).

**Table 1 jpm-12-01479-t001:** Mean number of bowel movements and mean BSS values. No values for week 3 are available.

	Mean Number of Bowel Movements/Week		Mean BSS Value/Week	
Week	Mean	std. Deviation	Mean	std. Deviation
1	9.1	3.2	3.8	0.6
2	9.1	4	3.7	0.6
4	9.1	3.4	3.7	0.6
5	9.4	3	3.7	0.6
6	9.1	3.3	3.7	0.6
7	8.9	3.5	3.8	0.7
8	8.7	2.9	3.8	0.7
9	9	3.3	3.7	0.6

## Data Availability

Sequence data are publicly available at EMBL-EBI and scripts detailing the reported statistical analysis and output have been deposited at github (https://github.com/reider-si/ELDERGUT).

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
