# Peer review of "Short- and Long-Term Effects of a Prebiotic Intervention with Polyphenols Extracted from European Black Elderberry—Sustained Expansion of Akkermansia spp."

_jpm, 2022, doi:10.3390/jpm12091479_

Round 1

Reviewer 1 Report

Dear authors,

The manuscript with ID jpm-1892245 presents relevant information about using polyphenols extracted from black elderberry as prebiotics to treat abdominal dysbiosis. Your approach is of much interest and deserves to gain more attention as the results evidence these bioactive compounds influence and modulate the abundance in the microbial community, especially in terms of Akkermansia species. Still, some changes should be made to the present form of the manuscript, as some aspects must be clarified. Specific comments are mentioned in the attached document.

Sincerely yours,

Reviewer

Author Response

Dear Editor, Dear Reviewer,

we would like to thank you for evaluating and commenting on our manuscript. The points raised during the peer review process provide valuable input to further improve our work and accentuate the manuscript’s main points.

We have submitted a revised manuscript in .docx format with track changes and list all differences compared to the first version herein.

The manuscript with ID jpm-1892245 presents relevant information about using polyphenols extracted from black elderberry as prebiotics to treat abdominal dysbiosis. Your approach is of much interest and deserves to gain more attention as the results evidence these bioactive compounds influence and modulate the abundance in the microbial community, especially in terms of Akkermansia species. Still, some changes should be made to the present form of the manuscript, as some aspects must be clarified. Specific comments are mentioned in the attached document.

We would like to thank the reviewer for their appreciative words regarding our work as well as for providing insight and a detailed list of corrections in the attached document. We comment on each point raised within the attached document in the following paragraphs:

„Please remove the brand name from the Abstract section…“
We have removed the brand name of the prebiotic preparation administered in the study from the abstract. It is still retained in the Methods section.

„The term dysbiosis should be narrowly defined…“
We thank the reviewer for making this important point. Indeed, the concept of dysbiosis is a problematic one within microbiome science as it is not clearly defined and often used without defining it. Furthermore, it does serve to explain causal relationships of specific microbial changes with observed outcomes. We are particularly grateful to the reviewer for pointing us towards the article by Brüssow (Microbial Biotechnology 2019, doi: 10.1111/1751-7915.13479) which discusses these issues in depth and provides a comprehensive overview of this topic. Thus, we have adapted the first paragraph of the introduction. It now reads as follows: “Although to date no distinct pattern of a healthy microbiota has emerged, certain alterations of microbiota composition and function are described as markers of disease-associated states. Collectively, these changes are summarized as dysbiosis, although a clear definition of the term is lacking and it has grown to be regarded increasingly unhelpful by experts in the field [1]. Nevertheless, loss of microbial diversity, a reduction in the number of beneficial microbes and an expansion of pathobionts are regarded as hallmarks of dysbiosis [2, 3].”

  1. Brüssow, H., Problems with the concept of gut microbiota dysbiosis. Microbial Biotechnology, 2020. 13(2): p. 423-434.
  2. Eckburg, P.B., et al., Diversity of the human intestinal microbial flora. Science, 2005. 308(5728): p. 1635-8.
  3. Human Microbiome Project, C., Structure, function and diversity of the healthy human microbiome. Nature, 2012. 486(7402): p. 207-14.

„Please give an appropriate reference at the end of the paragraph. Also please indicate the international authority with the regulation or legislation which specifies that the polyphenols fulfill the criteria of being categorized as prebiotics
We agree with the reviewer, that this is a highly relevant question. However, scientific classification as a potential prebiotic has to be clearly separated from regulatory approval of a specific compound as a prebiotic. It is difficult to formulate a statement on the legal situation which is currently very heterogeneous and dynamic with conflicting positions within member states of the European Union. This is also reflected in the situation regarding probiotics, which are likewise not uniformly regulated in the EU and changes are implemented continuously. In general, legislation lags behind scientific knowledge by several years. We thus have rephrased the sentence in question to better reflect the situation and distinguish more clearly between the scientific concept of a prebiotic and its approval by the component regulatory authorities. It thus reads now: “… the ISAPP has pointed out that some plant polyphenols might also fulfill these criteria and therefore might be considered prebiotics. However, this is currently not reflected in the legal and regulatory status of these compounds within the European Union”.

“Please add appropriate references” (page 2, regarding detectability of polyphenol metabolites in the urine)
We apologize for omitting the relevant reference and have added the work by Marin et al (doi.org/10.1155/2015/905215)

“Please use the following reference mentioning the polyphenols duplicity”
We would like to thank the reviewer for pointing us towards additional literature supporting this concept and have included the provided reference in the manuscript

“Please indicate appropriate reference to support your statement” (page 3, “Additional polyphenols present in…”)
We agree with the reviewer, that a reference specifically for that statement was missing from the paragraph and have added the work of Lee and Finn (J.Sci.Food Agric., 2007, 87:2665-2675).

“Please indicate the inclusion/exclusion criteria of the participants in the study. Please mention how many females were involved, the age range of participants and the logged digestive symptoms of the participants. In addition, if the study took place during COVID-19, you should also mention if the patients were or were not positive for the virus”
A paragraph on in- and exclusion criteria was added to the subsection. It reads: “Participants could be either male or female and had to be 18 to 50 years old. These in-dividuals had to report a history of vaginal delivery and being breast-fed for at least 3 months after birth. Exclusion criteria were recent antibiotic treatment within 3 months of the beginning of the study, a history of gastrointestinal disease or other acute or chronic medical conditions with the potential to affect the intestinal microbiome, a history of major abdominal surgery (appendectomy excluded), baseline consumption of dietary sup-plements or probiotics, a strictly vegan diet, relevant pathologic abnormalities in baseline clinical biochemistry tests and a known allergy to black elderberry.”
We have restructured the Materials & Methods section as well as the first part of the results (previously including results on the trial cohorts composition) and now present all information within the trial cohort subsection of the Materials & Methods section as requested. Briefly, the initial ratio of male and female participants was equal (n = 15 per sex). However, exclusions from the trial cohort due to newly diagnosed celiac disease affected 2 female participants, thus the final number of females in the reported cohort is 13. Mean age and age range are now reported in the same subsection of the manuscript (mean age 23.7 years +/- 3.05 years, range 19-31 years). Study subjects were asked for the presence of gastrointestinal symptoms including increase or decrease of appetite, the occurrence of flatulence and diarrhea, nausea, constipation and abdominal discomfort. Additionally, the number of bowel movements was recorded and summed on a weekly basis and fecal consistency was graded using the Bristol Stool Scale. The assessment of these patient reported outcomes is stated within the ‘Trial design and study cohort’ subsection of the Materials & Methods section of the revised manuscript.
The trial was conducted in autumn 2021 at a time of high prevalence of SARS-CoV2 in Austria. Thus, we have added the following paragraph to the ‘Trial design and study cohort’ subsection: “During conduction of the trial, 2 male participants contracted SARS-CoV2 during week 5 of the study period. The disease course was mild and the infection did not result in exclusion from the study or patients not providing samples.”

Please provide the producer’s location (city, country).
We have provided the location details for IPRONA AG/SPA which resides in Lana, province of Bolzano, Italy.

If available, please mention the classes/types of predominant anthocyanins and polyphenols.
This information is available and has been added with paracenteses to the words anthocyanins (primarily cyanidin 3-sambubioside-5-glucoside, cyanidin 3,5-diglucoside, cyanidin 3-sambubioside, cyanidin 3-glucosid) and polyphenols (primarily [1]chlorogenic acids, rutin, isoquercitrin).

“Please mention some examples of pre-dominant species of a-diversity”

We agree with the reviewer, that information on overally pre-dominant bacterial taxa is presented insufficiently in the manuscript draft. We have added a paragraph listing examples of the most prevalent bacterial OTUs and their mapping to genus level taxonomic information. It reads: “The most prevalent OTUs were mapped to the genus Bacteroides (phylum Bacteroidetes) and Faecalibacterium (phylum Firmicutes). Additionally, other genera from the phylum Firmicutes were highly abundant both at baseline and throughout the study period (i.e. Agathobacter, Ruminococcus, Roseburia). This was also observed for the genenera Bifidobacterium (phylum Actinobacteria) and Akkermansia (phylum Verrucomicrobia).“ However, as these taxa are among the most abundant throught the study period, it has to be noted, that they do not fully explain observed differences in measures of a-diversity.

“That aspects should be mentioned at Trial design and study cohort” & “Please use this data in the Trial design and study cohort”

We are grateful for this suggestion, which presents this information in a more accessible way. We have restructured the manuscript as suggested and moved this information to the relevant subsection of Materials and Methods.

“Please capitalize the words for abbreviations”
We thank the reviewer for this suggestion and have changed the spelling as suggested.,

“A conclusion section must be added highlighting the main findings of the study”
In our view, a conclusion is implicitly included in the final paragraph of the discussion. We have moved this paragraph to a new section under the heading “5. Conclusion” to comply with the reviewer’s request.

Due to the additional references provided and requested by the reviewers, there have been changes to the reference list, the details of which are evident from the revised manuscript file.

Furthermore, we noted that significance indicators were missing from Fig.3 A and B. We have added these to the figure, replaced the figure in the revised version of the manuscript and uploaded a new and updated version of the high-resolution file for this figure to the online submission system.

As the EMBL-ENI repository containing raw read data relevant to the findings reported in the manuscript has been made public, we have included the identifier of the repository in the Data availability statement. The data availability statement thus reads: “Sequence data are publicly available at EMBL-EBI (PRJEB55157) and scripts detailing the reported statistical analysis and output have been deposited at github (https://github.com/reider-si/ELDERGUT).

We are confident that the changes to the manuscript presented in this point-to-point reply address all of the reviewer’s concerns.

Sincerely,

Simon Reider & Alexander Moschen

Reviewer 2 Report

This is an excellent study with significant insights for the field. The observations on the resilience of the microbiome at key points in the intervention are interesting. Also the point made of using young healthy vs. dysbiotic individuals for study is important. The study design was excellent and is consistent with observations on other prebiotics such as xylo-oligosaccharides. However, polyphenols are underappreciated generally as prebiotics so these observations are significant to the general nutrition community. 

Author Response

Dear editor, dear reviewer,

We appreciate the reviewer’s approving comments regarding our study and would like to express our gratitude for reviewing the manuscript. We agree that polyphenols and their prebiotic properties are currently not receiving the attention that they deserve and hope to do our part to change this with our study.

Due to the additional references provided and requested by the reviewers, there have been changes to the reference list, the details of which are evident from the revised manuscript file.

Furthermore, we noted that significance indicators were missing from Fig.3 A and B. We have added these to the figure, replaced the figure in the revised version of the manuscript and uploaded a new and updated version of the high-resolution file for this figure to the online submission system.

As the EMBL-ENI repository containing raw read data relevant to the findings reported in the manuscript has been made public, we have included the identifier of the repository in the Data availability statement. The data availability statement thus reads: “Sequence data are publicly available at EMBL-EBI (PRJEB55157) and scripts detailing the reported statistical analysis and output have been deposited at github (https://github.com/reider-si/ELDERGUT).

We are confident that the changes to the manuscript presented in this point-to-point reply address all of the reviewer’s concerns.

Sincerely,

Simon Reider & Alexander Moschen

Round 2

Reviewer 1 Report

--